# CROSS3DREG: TOWARDS A LARGE-SCALE REAL-WORLD CROSS-SOURCE POINT CLOUD REGISTRATION BENCHMARK

## ABSTRACT

Cross-source point cloud registration, which aims to align point cloud data from different sensors, is a fundamental task in 3D vision. However, compared to the same-source point cloud registration, cross-source registration faces two core challenges: the lack of publicly available large-scale real-world datasets for training the deep registration models, and the inherent differences in point clouds captured by multiple sensors. The diverse patterns induced by sensors pose great challenges in robust and accurate point cloud feature extraction and matching, which negatively influence the registration accuracy. To advance research in this field, we construct Cross3DReg, the currently largest and real-world multi-modal cross-source point cloud registration dataset, which is collected by a rotating mechanical LiDAR and a hybrid semi-solid-state LiDAR, respectively. Moreover, we design an overlap-based cross-source registration framework, which utilizes unaligned images to predict the overlapping region between source and target point clouds, effectively filtering out redundant points in the irrelevant regions and significantly mitigating the interference caused by noise in non-overlapping areas. Then, a visual-geometric attention guided matching module is proposed to enhance the consistency of cross-source point cloud features by fusing image and geometric information to establish reliable correspondences and ultimately achieve accurate and robust registration. Extensive experiments show that our method achieves state-of-the-art registration performance. Our framework reduces the relative rotation error (RRE) and relative translation error (RTE) by $63.2\%$ and $40.2\%$, respectively, and improves the registration recall (RR) by $5.4\%$, which validates its effectiveness in achieving accurate cross-source point cloud registration.

## 1 INTRODUCTION

Cross-source point cloud registration Zhao et al. (2025a); Huang et al. (2021b) is a fundamental task in 3D vision, which plays an important role in robot Zhao et al. (2025b), autonomous driving Kim et al. (2025). The goal of cross-source point cloud registration is to align point clouds acquired from different sensors to construct complete 3D scenes Huang et al. (2023a) or estimate the robot location on maps Wang et al. (2025).

Compared to the same-source point cloud registration, advancements in cross-source registration are relatively slow for two main reasons. Firstly, there is a severe lack of public benchmark datasets that possess sufficient cross-source point cloud pairs. Existing public datasets, such as 3DCSR Huang et al. (2021b) and KITTI-CrossSource Xiong et al. (2024), exhibit notable limitations. 3DCSR Huang et al. (2021b) provides point clouds of Kinect-SFM indoor scenes. Its data scale is relatively small and insufficient for training deep registration models. The KITTI CrossSource dataset Xiong et al. (2024) consists of LiDAR scans and reconstructed point clouds from sequences using MonoRec Wimbauer et al. (2021). In these two cross-source point cloud datasets, the source or target point clouds are mainly synthesised with image sequences, not captured with real sensors. Secondly, as shown in Figure 1, point clouds scanned from different types of real scanners exhibit significant variance in data density and structural pattern. For example, point clouds captured by rotating LiDAR typically display sparse ring-like structures, whereas point clouds scanned by semi-solid-state LiDAR are often fan-shaped. In addition, point clouds acquired from different sensors

vary considerably in terms of noise levels, outlier distributions, and missing regions. These discrepancies can easily lead to a large number of mismatched features, which in turn severely degrade registration performance. Moreover, challenges from the real world, like inherent noise, outliers from capturing sensors and different data structure patterns, are not presented in currently available cross-source datasets. Facing these real-world cross-source point clouds, the accuracy of the most existing same-source point cloud registration methods Yu et al. (2023); Ren et al. (2024); Mu et al. (2024); Jiang et al. (2025) often deteriorates significantly due to different levels of noise and variance in density and structural patterns. Recent cross-source point cloud registration methods Zhao et al. (2025a); Xiong et al. (2024); Huang et al. (2023b; 2017a) are also proposed to improve registration accuracy. However, they are only designed for synthetic cross-source settings like Kinect-SFM, not real-world cross-source datasets.

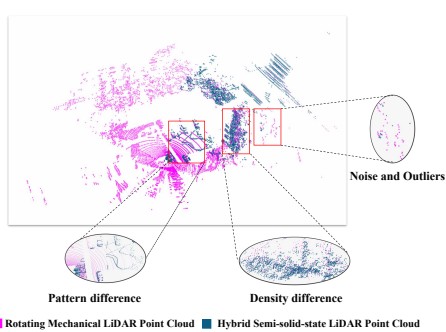

Figure 1: Challenges like differences in structural pattern and density, and realistic noise and outliers presented in real-world cross-source point clouds.

To address the above challenges, we first construct a large-scale real-world cross-source point cloud registration dataset named Cross3DReg. The source point clouds are captured by a hybrid semi-solid-state LiDAR, and the target point clouds are acquired with a 64-line rotating mechanical LiDAR. We also capture front views of the scenes from a roughly co-located view with the LiDAR. To mitigate interference from outliers and noise in point cloud matching, as well as the feature inconsistency arising from differences in cross-source point cloud density and structural pattern, we propose an overlap-based cross-source point cloud registration method by leveraging images to provide consistent visual clues. To this end, a dual-modal encoder is designed that effectively fuses image features with the coarse-grained geometric features of point clouds, which allows for predicting the probability of a 3D point located in the overlap region between source and target point clouds. In this way, we can effectively filter out redundant points and reduce noise out of the image view. After we predict the possible points locating in the overlap region between source and target point clouds, it is expected to establish correct correspondences among these points. However, as the source and target cross-source point clouds present different point patterns, distributions, resolutions and levels of noise and outliers, traditional geometric feature based matching methods are infeasible. Considering that correct correspondences within overlap region should have consistent features and the common image information is beneficial to enhance the feature consistency, we propose a visual-geometric attention-guided matching module to enhance the feature consistency between cross-source point clouds by fusing visual and geometric information adaptively with an attention mechanism, thereby establishing reliable correspondences. Please note that our method does not rely on calibration between cameras and LiDAR sensors. This is beneficial to avoid the inaccurate calibration caused by inevitable sensor vibration and the inherent extrinsic parameter drift. By leveraging images to enhance the consistency of point cloud features, our approach achieves accurate registration as long as the images are captured from a nearly co-located view, regardless of camera positions. This relaxed requirement for image input enhances the practical value of our approach. Experimental results also demonstrate the strong generalization capability of our method across images of varying quality and frames.

Our main contributions are listed below:

- To the best of our knowledge, Cross3DReg is the first large-scale real-world cross-source point cloud registration dataset. It includes $13,231$ point cloud pairs where different levels of noise, outliers, densities, and structural patterns are presented. Images showing common views between source and target point clouds are also collected. The dataset and code will be released.

- An overlap-based cross-source point cloud registration method is proposed to achieve accurate registration by predicting the overlap region with the help of unaligned images and ignoring redundant points and noise.

- To achieve accurate feature matching within the overlap region, a visual-geometric attention-guided matching module is proposed to fuse visual and geometric information adaptively with an attention mechanism, enhancing feature consistency of points within the overlap region between cross-source point clouds.

## 2 RELATED WORK

**Same-source point could registration**. Currently, same-source point cloud registration methods can be categorized into three classes. The first class is traditional iterative optimization-based approaches Besl & McKay (1992); Rusinkiewicz & Levoy (2001); Segal et al. (2009). However, these methods often suffer from a significant drop in registration accuracy when dealing with noisy or structurally complex scenes. The second class is correspondence-based point cloud registration methods Choy et al. (2019); Wang et al. (2022); Yu et al. (2023). Early approaches primarily rely on handcrafted feature descriptors Rusu et al. (2008; 2009) to establish point-wise correspondences. With the rapid development of deep learning in point cloud registration, a series of deep learning-based feature extraction models Bai et al. (2020); Ao et al. (2021) have been proposed, enabling more accurate and robust registration. Nevertheless, these methods still experience notable performance degradation when applied to regions with low overlap or a large number of outliers. To address these challenges, coarse-to-fine registration strategies Yu et al. (2021); Qin et al. (2023) have recently been adopted, showing accurate registration results under conditions of low overlap and high noise levels. The third class is end-to-end point cloud registration methods Xu et al. (2021); Zhang et al. (2022b); Lu et al. (2019). Unlike conventional two-stage registration frameworks, end-to-end approaches directly utilise deep neural networks to predict rigid transformations between point clouds without explicitly establishing point correspondences, thereby improving overall registration efficiency. Moreover, since raw point cloud data contains only geometric information, recent multimodal point cloud registration methods Zhang et al. (2022a); Xu et al. (2024; 2025) attempt to enhance feature discriminability by incorporating additional modalities (like colour, semantics, texture), allowing for more reliable correspondences. However, they all rely on explicit camera calibration to achieve geometric correspondence between pixel and point cloud, making them difficult to adapt to real-world scenarios.

**Cross-source point cloud registration**. The core challenge in cross-source point cloud registration lies in addressing the significant discrepancies introduced by different types of sensors. Compared to the same-source registration, cross-source point clouds exhibit a gap in density and pattern distribution, and are more susceptible to outliers. Traditional methods Huang et al. (2017b; 2019) are not designed to cope with these problems. To address the specific challenges posed by cross-source data, in recent years, mainstream cross-source point cloud registration methods Ma et al. (2024); Xiong et al. (2024); Zhao et al. (2025a) widely adopt a coarse-to-fine strategy. Correspondences are established by learning consistent deep features between the cross-source point clouds, thereby achieving robust registration. However, the advancement of this field is constrained by limited datasets; the efficacy of current approaches Zhao et al. (2024; 2025a) has primarily been validated on cross-source datasets where the source/target point clouds are synthesised with images, not real-world cross-source datasets. Therefore, developing cross-source point cloud registration data using real-world sensors represents a key future research objective in this area.

## 3 THE CROSS3DREG DATASET

To facilitate the development of cross-source point cloud registration, we introduce a large-scale real-world Cross3DReg dataset that contains $13,231$ point cloud pairs captured by a Rotating mechanical LiDAR and a Hybrid semi-solid-state LiDAR. The images are collected at co-located view with the Hybrid semi-solid-state LiDAR. And only one sensor is activated during a single data collection session. Table 1 shows the differences of the Cross3DReg compared with other cross-source datasets. More details about the capturing equipment, scanning process and the overlap ratio of Cross3DReg dataset, please refer to the *Appendix* A.1.

| Dataset | Scenes | Sensors | Pairs | Img. | Po. | Rno. | Rdp. | Dens. | Open. |
|---|---|---|---|---|---|---|---|---|---|
| 3DCSR | Indoors | Kinect, LiDAR, SFM | 202 | × | ✓ | × | × | ✓ | ✓ |
| KITTI-CrossSource | outdoors | LiDAR, SFM | 2006 | × | ✓ | × | × | ✓ | ✓ |
| Cross3DReg | outdoors | RL,HL | 13231 | ✓ | ✓ | ✓ | ✓ | ✓ | ✓ |

Table 1: The comparison of cross-source point cloud datasets. Img: RGB images. Po: Partial overlap. Rno: Real-world noise and outliers. Rdp: Real difference of structural pattern. Dens: Density difference. Open: Open source. RL: Rotating mechanical LiDAR. HL: Hybrid semi-solid-state LiDAR.

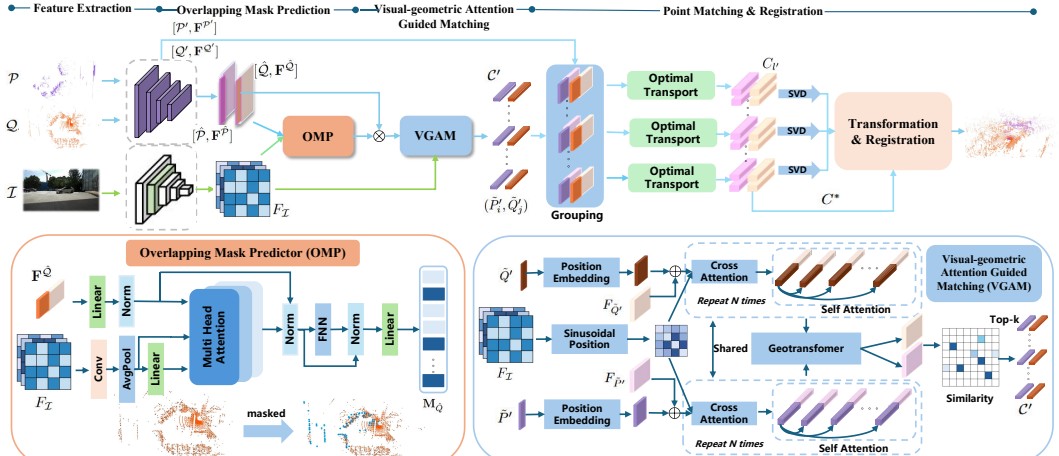

Figure 2: Our Cross3DReg registration method consists of four parts. We first encode the source and target point clouds, $\mathcal{P}$ and $\mathcal{Q}$, and an unaligned image $\mathcal{I}$, to extract superpoints and corresponding features. Based on the OMP module, image features are then fused with each point cloud's features to predict if superpoints are within the image view with the predicted binary mask vector $\mathbf{M}_{\hat{Q}}$ for the target (and $\mathbf{M}_{\hat{P}}$ for the source). With these superpoint sets of the source and target, the VGAM module establishes superpoint correspondences $\mathcal{C}'$ between source and target point clouds based on the visually enhanced features. Finally, these established correspondences are propagated to the original dense point clouds to generate final correspondences $\mathcal{C}_{l'}$, which are fed into a pose estimator to compute the transformation matrix.

## 4 METHOD

**Problem Statement**. The cross-source point cloud registration is formulated as follows. Given the source point cloud $\mathcal{P} = \{\mathbf{p}_i \in \mathbb{R}^3 | i = 1, \ldots, N\}$ and target point cloud $\mathcal{Q} = \{\mathbf{q}_j \in \mathbb{R}^3 | j = 1, \ldots, M\}$, we estimate an optimal rigid transformation matrix $\mathcal{T} = \{\mathbf{R}, \mathbf{t}\}$ to align $\mathcal{P}$ and $\mathcal{Q}$, where $\mathbf{R} \in SO(3)$ is the rotation matrix and $\mathbf{t} \in \mathbb{R}^3$ is the translation vector. The optimisation goal can be formulated as:

$$\mathcal{T} = \arg\min_{R,t} \sum_{(p_i, q_j) \in C^*} \|\mathbf{R}\mathbf{p}_i + \mathbf{t} - \mathbf{q}_j\|_2, \tag{1}$$

where $\mathcal{C}^*$ represents the correspondences between $\mathcal{P}$ and $\mathcal{Q}$ and the $\|.\|_2$ denotes the Euclidean distance.

For accurate cross-source registration, we propose an overlap-based cross-source point registration method to establish the correspondence between point clouds. As shown in Figure 2, the framework contains four phases: 1) Feature extraction. Features of Image $\mathcal{I}$ and point clouds $\mathcal{P}$ and $\mathcal{Q}$ are extracted through a two-branch network. The point cloud branch downsamples the original point clouds to acquire superpoints $\hat{\mathcal{P}}$ and $\hat{\mathcal{Q}}$. The corresponding features $\mathbf{F}^{\hat{\mathcal{P}}}$ and $\mathbf{F}^{\hat{\mathcal{Q}}}$ are also extracted; 2) Overlapping Mask Predictor (OMP). Although images are not aligned with point clouds, they contain the common views of the source and target point clouds. Thus, the coarse-grained superpoint features are firstly fused with image features to predict the mask of superpoints that indicates if the superpoint is within the image view. Both the overlap masks of the source and target point clouds are predicted. The following matching step is only performed between overlapping masks. 3) Visual-Geometric Attention guided superpoint Matching (VGAM). With the overlapping masks, visually

enhanced superpoint features are utilised for similarity computation, and robust coarse-grained correspondences are established between the source and target. 4) Point matching and registration. Based on the superpoint matching results, precise point-level correspondences are obtained by local point cloud feature matching. Based on the point correspondences, transformations are estimated using the pose estimators.

## 4.1 FEATURE EXTRACTION

Due to the significant density variations and a large number of points, we first preprocess the raw point clouds using a voxel-based downsampling method (voxel size = 0.25) before feeding them into the feature extractor. The downsampled source point cloud $\mathcal{P}$ and the target point cloud $\mathcal{Q}$ are then put into the KPConv-FPN backbone network Thomas et al. (2019), which can extract the point cloud features at different scales. At the coarsest scale, we obtain the superpoint sets $\hat{\mathcal{P}}$ and $\hat{\mathcal{Q}}$. Their corresponding features are $\mathbf{F}^{\hat{\mathcal{P}}} \in \mathbb{R}^{|\hat{P}| \times \hat{d}}$ and $\mathbf{F}^{\hat{\mathcal{Q}}} \in \mathbb{R}^{|\hat{Q}| \times \hat{d}}$, $\hat{d}$ is the feature dimension of superpoints at the coarsest level. Points at the densest level are denoted as $\mathcal{P}'$, $\mathcal{Q}'$ and its corresponding features are $\mathbf{F}^{\mathcal{P}'} \in \mathbb{R}^{|\mathcal{P}'| \times d'}$ and $\mathbf{F}^{\mathcal{Q}'} \in \mathbb{R}^{|\mathcal{Q}'| \times d'}$, $d'$ is the corresponding feature dimension.

For image feature extraction, we employ a U-Net backbone network with residual connections to process the intermediate unaligned images. Given an input image $\mathcal{I} \in \mathbb{R}^{H \times W}$, its feature is represented as $F_{\mathcal{I}} \in \mathbb{R}^{H \times W \times d}$.

## 4.2 OVERLAPPING MASK PREDICTOR MODULE

As various point densities, prevalent noise, outliers and different structural patterns exist in the cross-source point cloud registration, traditional point features-based overlapping region estimation is inaccurate. Since the captured images in our dataset contain the common views, we can utilise these images to identify possible points that are located in the overlap region between the source and target point clouds. This is also beneficial to filter redundant points out of the image view.

Therefore, we propose an overlapping mask prediction module based on misaligned images. We define this problem as a binary regression task, fusing image information to directly perform linear regression for the mask of each superpoint. Since the camera calibration information is unavailable, images and point clouds are unaligned. We first perform image and point cloud feature dimension alignment. The image features $\mathbf{F}_{\mathcal{I}} \in \mathbb{R}^{H \times W \times d}$ and superpoint features $\mathbf{F}^{\hat{\mathcal{Q}}} \in \mathbb{R}^{|\hat{Q}| \times \hat{d}}$ are mapped to a unified dimensional space by linear projection to obtain aligned features. The aligned image features and the superpoint features of point clouds are represented as $\widehat{\mathbf{F}}^{\hat{\mathcal{Q}}}$ and $\widehat{\mathbf{F}}_{\mathcal{I}}$. Then, we use the multi-attention mechanism Vaswani et al. (2017) to carry out the cross-modal feature fusion. Finally, the fused features are processed by a residual feed-forward network (FFN) with layer normalization. The overlap probability of superpoints is output via an MLP net. Taking the prediction of the overlap mask for the target point cloud $\mathbf{Q}$ as an example, the process can be formalized as follows.

$$\mathbf{F}_{fuse} = MultiHeadAttn(\widehat{\mathbf{F}}^{\hat{\mathcal{Q}}}, \widehat{\mathbf{F}}_{\mathcal{I}}). \tag{2}$$

Given the fused image and point cloud features, the probability of each superpoint belonging to the overlap region, denoted as $\mathbf{P}^{\hat{Q}}_{overlap}$, can be estimated as:

$$\mathbf{P}^{\hat{Q}}_{overlap} = \sigma(MLP(\mathbf{F}_{fuse} + (FFN(\mathbf{F}_{fuse} + \widehat{F}^{\hat{\mathcal{Q}}})))), \tag{3}$$

$$\mathbf{M}_{\hat{Q}_i} = \begin{cases} 1 & \text{if } \mathbf{P}^{\hat{Q}_i}_{overlap} > \lambda \\ 0 & \text{otherwise} \end{cases}, \tag{4}$$

where $\mathbf{M}_{\hat{Q}} \in \{0, 1\}$ is a binary mask vector, $\sigma$ represents the sigmoid function, and the $\lambda$ denotes the confidence threshold, default is $0.5$.

## 4.3 VISUAL-GEOMETRIC ATTENTION GUIDED MATCHING

To effectively incorporate visual context into geometric features for point cloud registration, we introduce the visual-geometric attention guided matching module to leverage both visual and geometric information to enhance the consistency of superpoint features between point cloud of different sources. The core of our approach is a two-stage attention mechanism. First, the visual cross-attention mechanism fuses visual context from image features into the superpoint features. Subsequently, a geometric self-attention mechanism Qin et al. (2023) refines these fused features to capture global geometric relationships within the point cloud.

Let's take the target point cloud $\mathcal{Q}$ as an example. We first use a predicted overlap mask $\mathbf{M}_{\hat{Q}}$ to select a subset of superpoints $\tilde{\mathbf{Q}}'$ and their corresponding features $\mathbf{F}_{\tilde{Q}'}$. The corresponding image features $\mathbf{F}_I$ are flattened into a vector. To provide the attention mechanism with spatial awareness, we introduce positional encoding, $\mathbf{F}_{pos}^{\tilde{Q}'}$ for superpoints and $\mathbf{F}_{pos}^{I}$ for image pixels.

The superpoint features, image features, and their respective positional encodings are projected into Query ($\mathbf{Q}_c$), Key ($\mathbf{K}_c$), and Value ($\mathbf{V}_c$) spaces using learnable linear matrices. The positional encodings are also projected to generate point cloud positional embeddings ($\mathbf{E}_c$) and image positional embeddings ($\mathbf{G}_c$). The process is formulated as:

$$\mathbf{Q}_c = \mathbf{F}_{\tilde{Q}'}\mathbf{W}_{Q_c}, \quad \mathbf{K}_c = \mathbf{F}_I\mathbf{W}_{K_c}, \quad \mathbf{V}_c = \mathbf{F}_I\mathbf{W}_{V_c}, \tag{5}$$

$$\mathbf{E}_c = \mathbf{F}_{pos}^{\tilde{Q}'}\mathbf{W}_{E_c}, \quad \mathbf{G}_c = \mathbf{F}_{pos}^{I}\mathbf{W}_{G_c}, \tag{6}$$

where $\mathbf{W}_{Q_c}, \mathbf{W}_{K_c}, \mathbf{W}_{V_c}$ are learnable projection matrices for the Query, Key, and Value, and $\mathbf{W}_{E_c}, \mathbf{W}_{G_c}$ are the projection matrices for their respective positional embeddings.

By integrating content and position information, we compute the cross-attention scores. These scores weigh the aggregation of the Value and image positional embeddings. The superpoint features are then updated via a residual connection:

$$\mathbf{Scores}_c = softmax\left(\frac{(\mathbf{Q}_c + \mathbf{E}_c)(\mathbf{K}_c + \mathbf{G}_c)^T}{\sqrt{d'}}\right), \tag{7}$$

$$\mathbf{F}'_{\tilde{Q}'} = \mathbf{Scores}_c(\mathbf{V}_c + \mathbf{G}_c) + \mathbf{F}_{\tilde{Q}'}, \tag{8}$$

where $\mathbf{F}'_{\tilde{Q}'}$ denotes the updated superpoint features enriched with visual context information.

To enhance the global structural integrity of the features and mitigate potential noise, we employ a self-attention mechanism. This step promotes information propagation across entire point clouds. The visually-enhanced features $\mathbf{F}'_{\tilde{Q}'}$ are linearly projected into a new set of Query ($\mathbf{Q}_s$), Key ($\mathbf{K}_s$), and Value ($\mathbf{V}_s$). Self-attention weights are then computed to update the features in a residual manner:

$$\mathbf{Scores}_s = softmax\left(\frac{\mathbf{Q}_s\mathbf{K}_s^T}{\sqrt{d'}}\right), \tag{9}$$

$$\mathbf{F}''_{\tilde{Q}'} = \mathbf{Scores}_s\mathbf{V}_s + \mathbf{F}'_{\tilde{Q}'}. \tag{10}$$

Finally, combined with the geometric self-attention mechanism, we can maximize the descriptive power of the final features. After the feature enhancement process, we can obtain highly discriminative superpoint features, $\bar{\mathbf{F}}_{\tilde{P}'}$ and $\bar{\mathbf{F}}_{\tilde{Q}'}$, for the source and target point clouds, respectively. We then construct a feature similarity matrix $\mathbf{Z}'$ by the following formulation:

$$\mathbf{Z}'_{ij} = \exp\left(-\|\bar{\mathbf{F}}_{\tilde{P}'_i} - \bar{\mathbf{F}}_{\tilde{Q}'_j}\|_2^2\right), \tag{11}$$

where $\mathbf{Z}'_{ij}$ measures the similarity between the $i$-th source superpoint $\tilde{\mathbf{P}}'_i$ and the $j$-th target superpoint $\tilde{\mathbf{Q}}'_j$. Finally, we apply dual normalization Rocco et al. (2018); Sun et al. (2021) to the similarity matrix $\mathbf{Z}'$ and select the top-K entries with the highest scores to form the final set of superpoint correspondences $\mathcal{C}' = \{(\tilde{\mathbf{P}}'_i, \tilde{\mathbf{Q}}'_j)|\tilde{\mathbf{P}}'_i \in \hat{\mathcal{P}}, \tilde{\mathbf{Q}}'_j \in \hat{\mathcal{Q}}\}$.

## 4.4 POINT MATCHING AND REGISTRATION

After obtaining the superpoint correspondences in the overlapping regions, we employ a point-to-node grouping strategy Yu et al. (2021) to further establish correspondences between the dense points. The core idea of this strategy is to assign dense points to their nearest neighboring superpoints based on spatial distance. Specifically, for a matched superpoint pair $(\tilde{\mathbf{P}}'_i, \tilde{\mathbf{Q}}'_j)$, we denote their corresponding dense point groups as $\mathbf{G}_{\tilde{P}'_i}$ and $\mathbf{G}_{\tilde{Q}'_j}$, and their feature groups as $\mathbf{G}_i^{\mathbf{F}^{\mathcal{P}'}}$ and $\mathbf{G}_j^{\mathbf{F}^{\mathcal{Q}'}}$, respectively. Based on the superpoint correspondence $(\tilde{\mathbf{P}}'_i, \tilde{\mathbf{Q}}'_j)$, we compute the similarity matrix between the feature groups $\mathbf{G}_i^{\mathbf{F}^{\mathcal{P}'}}$ and $\mathbf{G}_j^{\mathbf{F}^{\mathcal{Q}'}}$ as $\mathbf{S}_{l'} = \frac{\mathbf{G}_i^{\mathcal{P}'}(\mathbf{G}_j^{\mathbf{F}^{\mathcal{Q}'}})^T}{\tilde{d}}$, where $\tilde{d}$ denotes the feature dimension. To enhance matching robustness, we adopt the method from Sarlin et al. (2020) by adding slack terms, controlled by a learnable parameter $\alpha$, to the last row and column of the similarity matrix $\hat{\mathbf{S}}'$. Subsequently, we apply the Sinkhorn algorithm to find the optimal matching. After removing the slack terms, we select the top $K'$ matching pairs with the highest confidence scores to establish the group-level dense point correspondences $C_{l'}$. Finally, by aggregating the correspondences from all groups, we obtain the global correspondences $\mathcal{C}^* = \bigcup_{l'=1}^{|C'|} \mathcal{C}_{l'}$. Based on the correspondences, we employ the LGR estimator Qin et al. (2023) to accurately estimate the transformation matrix. The pseudocode of the proposed method is shown in *Appendix* A.3.

## 4.5 LOSS FUNCTION

Our loss function is composed of three components and can be expressed as: $\mathcal{L}_{total} = \mathcal{L}_{coarse} + \mathcal{L}_{fine} + \mathcal{L}_{mask}$. Here, following the framework of GeoTrans Qin et al. (2023), $\mathcal{L}_{coarse}$ and $\mathcal{L}_{fine}$ are the losses supervising the coarse-grained (superpoint) and fine-grained (point-level) matching, respectively. For $L_{mask}$, we employ the Focal Loss Lin et al. (2017). For further details regarding the loss function, please refer to the *Appendix* A.2.

## 5 EXPERIMENTS

**Metrics**. We use the following metrics to evaluate methods: *Relative Rotation Error* (**RRE**), *Relative Translation Error* (**RTE**), *Registration Recall* (**RR**), and *Inlier Ratio* (**IR**). For the Cross3DReg dataset, the threshold for RR is defined as RRE $< 2°$ and RTE $< 0.5$ m. The IR evaluates the quality of the point matching by calculating the proportion of corresponding points whose distances under the true transformation are below the threshold of 1.0m.

**Implementation Details**. All experiments are implemented based on the PyTorch framework and trained on the NVIDIA RTX A6000 GPU with the following key parameters: initial learning rate is $10^{-4}$; Batch size is 1, and the weight decay is $10^{-6}$. Our model is trained with Adam optimizer for 20 epochs.

## 5.1 QUANTITATIVE COMPARISON

To validate the effectiveness of the proposed method and evaluate its registration performance on the Cross3DReg dataset, we conduct a comparative experiment with several state-of-the-art point cloud registration approaches, as summarized in Table 2. The selected methods cover a diverse spectrum of methodologies, including: a traditional iterative optimization technique (ICP Besl & McKay (1992)); correspondence-based approaches (FCGF Choy et al. (2019), Omnet Xu et al. (2021), Predator Huang et al. (2021a),CoFiNet Yu et al. (2021), RoiTr Yu et al. (2023), GeoTrans Qin et al. (2023)). In addition, we compare with VRHCF Zhao et al. (2024), a cross-source point cloud registration method based on feature learning, as well as the multimodal registration method IMFNET Huang et al. (2022). The results show that our approach significantly outperforms the state-of-the-art methods in all evaluation metrics. In terms of registration accuracy, the lowest RRE $= 6.68°$ and RTE $= 1.01$m are achieved, which reduces the rotation and translation error by $63.2\%$ and $40.2\%$, respectively. Our method also achieves the highest RR metrics, which is $5.4\%$ higher than the state-of-the-art GeoTrans.

| | Methods | RRE(°) ↓ | RTE(m)↓ | RR(%) ↑ |
|---|---|---|---|---|
| Same-source | ICP | 94.69 | 9.30 | 0.0 |
| | FCGF | 94.70 | 9.00 | 0.0 |
| | Omnet | 95.01 | 11.20 | 0.0 |
| | Predator | 100.81 | 30.37 | 0.0 |
| | CoFiNet | 99.34 | 16.97 | 0.0 |
| | RoITr | 16.79 | 3.10 | 43.1 |
| | IMFNET | 94.8 | 9.07 | 0.0 |
| | GeoTrans | 18.15 | 1.69 | 81.7 |
| Cross-source | VRHCF | 110.62 | 16.35 | 0.0 |
| | Cross3DReg (Ours) | **6.68** | **1.01** | **87.1** |

Table 2: The comparison of the same- and cross-source registration methods on Cross3DReg.

Additionally, while methods such as FCGF, Omnet, Predator, IMFNET, and CoFiNet achieve strong performance on same-source point clouds, their Registration Recall (RR) drops to nearly zero in cross-source scenarios. This indicates the inherent difficulty for same-source registration methods in handling the cross-source cases where substantial point pattern variations and significant noise interference are prevalent. The keypoint-based methods like FCGF, Predator can hardly acquire correct point correspondences only based on the geometric features as geometry cannot remain consistent in cross-source point clouds. The coarse-to-fine methods, like CoFiNet, cannot achieve successful registration either. The accuracy of the initial super-point matching is pivotal for coarse-to-fine methods. However, under the influence of point cloud discrepancies and noise, super-point feature extraction becomes unreliable without the guidance of consistent information. After consistent geometric structural information, like angles and distances between super-points or rotation-invariant features, is fused in geometric features to enhance the consistency, GeoTransformer and RoITr methods can achieve more accurate registration. Notably, we also observe that IMFNET's global fusion approach is ineffective for cross-source registration. Failing to capture pixel-to-point correspondence, the method leads to feature mismatching and subsequent noise injection.

For the current cross-source VRHCF method, a spherical voxelization operation is used to resample the density-variant cross-source point clouds into evenly distributed point clouds and geometric fearures are then extracted. However, faced with our Cross3DReg, which fully presents real-world point uneven distribution, noise, outliers, and pattern difference, VRCHF cannot get evenly distributed point clouds only with a simple sampling method. The extracted geometric feature cannot maintain consistency and totally failed in our dataset.

| Estimator | Method | RRE(°) ↓ | RTE(m)↓ | RR(%) ↑ | IR(%) ↑ | Time (s)↓ | | |
|---|---|---|---|---|---|---|---|---|
| | | | | | | Model | Pose | Total |
| LGR | Geotrans | 18.148 | 1.690 | 81.7 | 60.1 | **0.186** | 0.032 | **0.218** |
| | RoiTr | 16.718 | 3.098 | 53.1 | 25.8 | 0.245 | 0.033 | 0.278 |
| | Ours | **6.683** | **1.010** | **87.1** | **70.3** | 0.252 | **0.031** | 0.283 |
| RANSAC-50k | RoiTr | 9.731 | 2.923 | 67.8 | 25.8 | 0.245 | 0.144 | 0.389 |
| | Predator | 100.81 | 30.37 | 0.0 | 0.25 | 0.354 | 0.564 | 0.918 |
| | CoFiNet | 99.34 | 16.97 | 0.0 | 0.13 | **0.073** | **0.026** | **0.099** |
| | Geotrans | 13.05 | 1.436 | 72.2 | 60.1 | 0.186 | 1.299 | 1.485 |
| | Ours | **8.150** | **1.135** | **78.6** | **70.3** | 0.252 | 1.353 | 1.605 |
| weighted SVD | Geotrans | 16.491 | 2.193 | **62.9** | 60.1 | **0.186** | **0.002** | **0.188** |
| | RoiTr | 54.21 | 16.54 | 0.2 | 25.8 | 0.245 | 0.003 | 0.248 |
| | Ours | **8.684** | **1.153** | 57.8 | **70.3** | 0.252 | 0.002 | 0.254 |

Table 3: Registration performance on the Cross3DReg dataset with different pose estimators. Model time refers to the time required for feature extraction, while pose time refers to the time required for transformation estimation.

To fully evaluate the correspondence of our cross-source point cloud registration method, we evaluate the registration accuracy with different estimators. As shown in Table 3, when combined with the LGR Qin et al. (2023) pose estimator, our method exhibits optimal performance in all key evaluation metrics. When we employ the RANSAC estimator, our method still outperforms with RRE = 8.150° and RTE = 1.135 m, significantly better than GeoTrans, Predator, and CoFiNet, which also

employ RANSAC. It is worth noting that both Predator and CoFiNet have an RR of $0.0\%$ in this setup, indicating that they fail to produce correspondences of sufficient quality for the RANSAC estimator to achieve accurate alignment. Finally, we perform the tests in the weighted SVD Besl & McKay (1992), our method still achieves accurate registration where RRE=$8.684°$, RTE=1.153 m, and registration recall reaches $57.8\%$. In contrast, the registration recall of RoiTr is only $0.2\%$. Furthermore, it is observed that despite our method achieving approximately $10\%$ higher inlier ratio and lower registration error compared to GeoTrans, there is a gap in registration recall. The core reason for this is not due to the low quality of our generated correspondences, but rather the insufficient robustness of the pose estimator. Specifically, weighted SVD, a least-squares solution, attempts to fit all given corresponding points. This makes it highly sensitive to the mismatched correspondences. Moreover, we also evaluate the computational efficiency of our approach. Despite requiring the processing of additional image information, our method achieves a balance between efficiency and performance. For the robustness of generalization of the proposed method, please refer to *Appendix A.5*.

## 5.2 QUALITATIVE COMPARISONS.

Besides the quantitative comparison, in Figure 3, we also present a qualitative comparison of our method against current baseline approaches in the selected three challenging scenes, which has significant differences in density and pattern. As can be observed, methods such as Predator and CoFiNet fail to achieve successful alignment in the three challenging scenarios. In contrast to RoiTr and GeoTransformer, our method attains the optimal registration performance, with results that are visually closest to the Ground Truth.

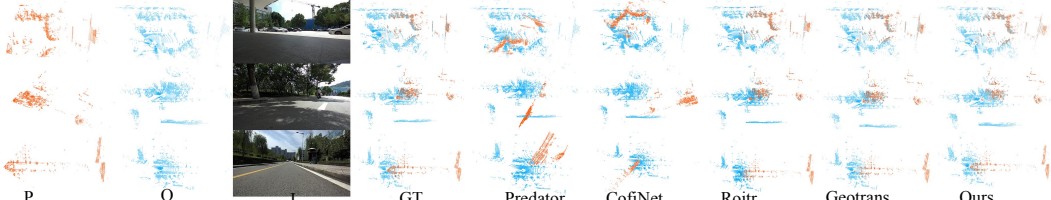

Figure 3: The qualitative comparison on Cross3DReg.

Figure 4 visualizes the point correspondences generated by different registration approaches. The qualitative comparison clearly reveals that, compared to current state-of-the-art methods, the correspondences extracted by our method are more precise and highly concentrated on the overlapping regions of the point clouds. In contrast, methods like Predator, CoFiNet, and Roit produce a substantial number of incorrect correspondences. We attribute this primarily to two factors: a significant decrease in feature consistency within the overlapping regions when faced with considerable density variations across point clouds, and severe interference in feature matching caused by prominent noise in the scenes. For more visual qualitative comparisons, please refer to *Appendix A.4*.

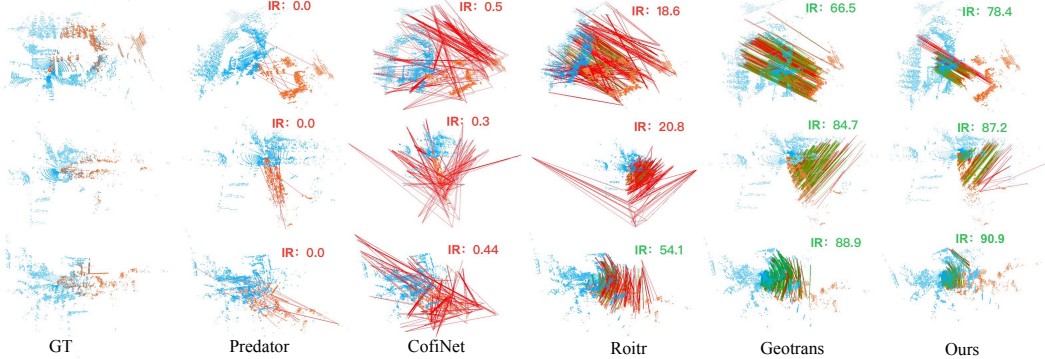

Figure 4: The visualization of correspondences on Cross3DReg.

Additionally, as shown in Figure 5, we provide a visual comparison of the overlapping regions between our proposed method and Omnet to show the necessity of using a common image to predict the overlap region. It can be observed that the point cloud overlap region estimated by our method aligns more closely with the ground truth, whereas Omnet's results exhibit noticeable deviations. This discrepancy arises because Omnet directly regresses the overlapping regions from the features of two point cloud frames. In cross-source outdoor scenarios, however, point clouds are typically structurally different and of different levels of noise and outliers, leading to low feature similarity between point clouds. As a result, the model struggles to accurately locate overlap regions, only relying on the geometric features. Incorrect predictions of overlap regions subsequently cause incorrect matches, resulting in registration failure.

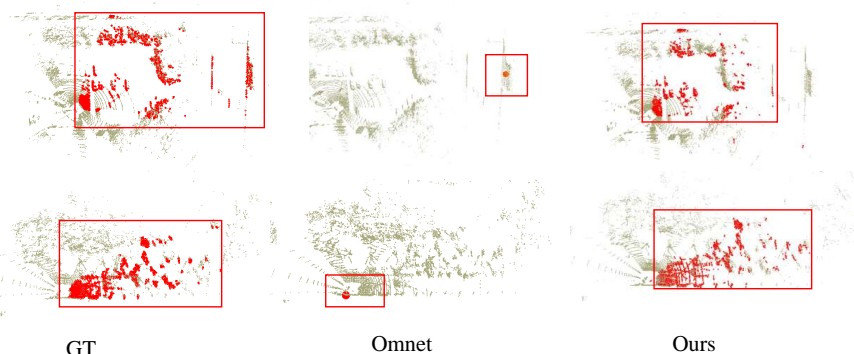

GT           Omnet           Ours

Figure 5: The comparison of overlap regions extracted with our method and Omnet.

## 5.3 ABLATION STUDIES

As shown in Table 4, we conduct ablation experiments on the Cross3DReg dataset to evaluate the effectiveness of modules proposed in the method.

| Method | RRE(°)↓ | RTE(m)↓ | RR(%)↑ | IR(%)↑ |
|---|---|---|---|---|
| (a) Geo self-attention w/o OMP | 18.184 | 1.690 | 81.7 | 60.1 |
| (b) VGAM w/o OMP | 17.812 | 1.592 | 82.1 | 60.3 |
| (c) OMP w/ Vanilla self-attention | 8.496 | 1.242 | 85.3 | 70.0 |
| (d) OMP w/ Geo self-attention | 8.371 | 1.287 | 86.7 | 70.1 |
| (e) OMP w/ VGAM(full) | **6.683** | **1.010** | **87.2** | **70.3** |

Table 4: The ablation study of each module of the proposed method.

Experiments are set up with five scenarios in comparison: (a) only the geometric self-attention module is used; (b) only the VGAM is used; (c) the OMP module is used with the vanilla attention module; (d) the OMP module is used with the geometric self-attention module; and (e) the complete Cross3DReg method. The comparison among (a), (b), (e) shows the OMP module can effectively mitigate the interference of redundant and noisy points, significantly improving the accuracy of cross-source point cloud alignment. The comparison of schemes (b), (c), (d), and (e) further shows that our visual-geometric feature attention guidance module improves the consistency of feature space among cross-source point clouds, enhancing the registration accuracy.

## 6 CONCLUSION

In this paper, we first propose a large-scale and real-world cross-source point cloud registration dataset, named Cross3DReg, to show different levels of noise, outliers, densities and structural patterns. An overlap-based cross-source point cloud registration method is then proposed to achieve accurate registration by predicting the overlap region with the help of unaligned images and ignoring redundant points and noise. Extensive experiments verify the challenges of our proposed dataset and the effectiveness of the proposed method.

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

## A   APPENDIX

### A.1   MORE DETAILS ABOUT CROSS3DREG DATASET

Here, we present the devices we used to collect the dataset, the scanning process and the overlap information between the source and target point clouds of Cross3DReg dataset in the following.

**(1) The capturing equipment.** As shown in Figure 6, our Cross3DReg dataset is collected by a custom-built Unmanned Ground Vehicle (UGV), equipped with a 64-beam rotating mechanical LiDAR, a hybrid semi-solid-state LiDAR, a RGB-D stereo camera (ZED camera) and RGB cameras. Due to the high quality of captured images, in our dataset, RGB images acquired with the left camera of ZED are used as the visual information and available to the public. The reasons for employing these sensors can be summerized as follows:

- According to our research, the primary representative cross-source point cloud datasets currently available are 3DCSRHuang et al. (2023b) and Kitti-CrossSourceXiong et al. (2024). 3DCSR comprises 202 pairs of indoor LiDAR and SFM cross-source point clouds, whilst Kitti-CrossSource contains 2006 pairs of outdoor LiDAR and SFM cross-source point clouds. It should be noted that at least one point cloud in these datasets is synthesised via SFM methods; the academic community currently lacks cross-source point cloud data entirely derived from real sensor acquisitions.

- From a sensor characteristics perspective, Roatating mechanical LiDAR boasts high precision and high cost, capable of capturing a complete 360° point cloud of the surroundings. It is commonly employed for high-precision map construction. Conversely, hybrid semi-solid-state LiDAR offers relatively lower accuracy at a more economical price point, yet can only acquire point cloud data within a limited field of view. It is typically mounted on small unmanned mobile devices. These two point cloud types exhibit distinct pattern differences, density variations, and varying degrees of noise. These characteristics fully reveal the core challenges inherent in cross-source point cloud registration.

**(2) The scanning process.** We design and execute 11 distinct collection routes across a university campus. These routes encompass typical campus environments, featuring a rich variety of dynamic and static elements, which include dynamic traffic participants (e.g., vehicles and pedestrians), static obstacles (e.g., road barriers), as well as infrastructure and natural landscapes (e.g., buildings and vegetation). Furthermore, to establish authentic cross-source acquisition conditions and maintain the independence of each modality, only one sensor is activated during a single data collection session. Table 1 shows the differences of the Cross3DReg compared with other cross-source datasets.

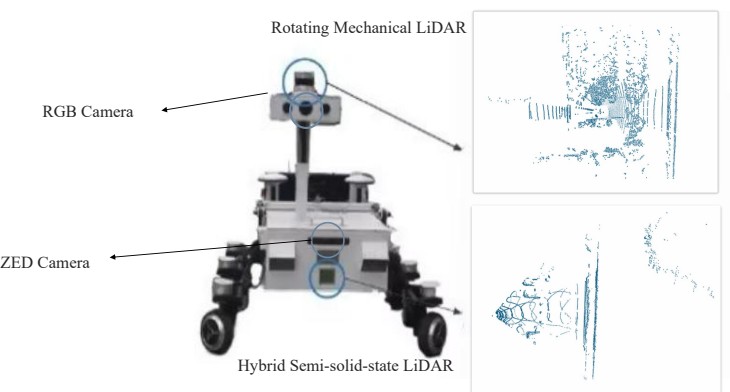

Figure 6: Data acquisition platform

**(3) Dataset information.** We split the dataset into training, validation, and test sets. Training data is drawn from routes 00-05 ($7,772$ pairs), validation data from routes 06–07 ($1,314$ pairs), and test data from routes 08–10 ($4,145$ pairs). For every source – target point cloud pair, the ground-truth label is the rigid transformation that precisely registers the source to the target. These transformations are deliberately varied, covering rotation errors from $8°$ to $180°$ and translation offsets from 3 m to 15 m. Figure 7 shows the statistics of the rotations and translations of Cross3DReg dataset. We also define the overlap ratio between the source and target point clouds as follows. Given two point clouds $\mathcal{P}_{src}$, $\mathcal{Q}_{ref}$, we compute the overlap ratio between two point clouds as follows:

$$\mathcal{P}_{src}^{'} = Trans(\mathcal{P}_{src}),  \tag{12}$$

$$\mathcal{O}_{ref \rightarrow src} = \frac{1}{|\mathcal{Q}_{ref}|} \sum_{i=1}^{|\mathcal{Q}_{ref}|} \mathbb{I}(min||q_i - p'||_2 < r),  \tag{13}$$

$$\mathcal{O}_{src \rightarrow ref} = \frac{1}{|\mathcal{P}_{src}^{'}|} \sum_{j=1}^{|\mathcal{P}_{src}^{'}|} \mathbb{I}(min||p_j - q||_2 < r),  \tag{14}$$

$$\mathcal{O} = min(\mathcal{O}_{ref \rightarrow src}, \mathcal{O}_{src \rightarrow ref}),  \tag{15}$$

where $\mathcal{Q}_{ref}^{'} = \{q_i \in \mathbb{R}^3 | i = 1, ..., m\}$; $\mathcal{P}_{src} = \{p_j \in \mathbb{R}^3 | j = 1, ..., n\}$; $Trans(.)$ represents the rigid transformation; $||.||_2$ is the Euclidean distance; $\mathbb{I}(.)$ is the indicator function; $r$ denotes the distance threshold. Based on a calculated average inter-point distance of 0.2m, we set the distance

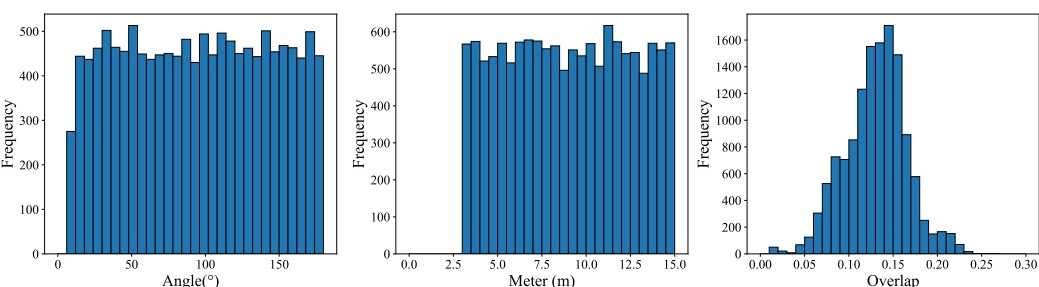

Figure 7: The statistics of the rotation angles, translation and the overlap ratio of Cross3DReg.

threshold $r$ to 0.2m. As shown in Figure 7, the cross3DReg dataset has a low point cloud overlap ratio of less than 30%.

Examples of the Cross3DReg dataset are visualized in Figure 8. Source and target point clouds are shown in the first two columns. It is obvious to see that the Cross3DReg dataset is very challenging where different point patterns, level of noise and outliers, and various densities are presented. Corresponding images and aligned point clouds are also presented in the last two columns.

### A.2 LOSS FUNCTIONS

Our total loss function $\mathcal{L}_{total}$ is composed of three parts. $\mathcal{L}_{total} = \mathcal{L}_{coarse} + \mathcal{L}_{fine} + \mathcal{L}_{mask}$. The definition of $\mathcal{L}_{coarse}$ and $\mathcal{L}_{fine}$ follows GeoTrans Qin et al. (2023). $\mathcal{L}_{mask}$ employs the Focal Loss Lin et al. (2017). Due to the lack of camera intrinsic and extrinsic parameters, which prevents the establishment of a projection relationship from the 3D point cloud to the 2D image, we adopt a mask generation strategy based on the point cloud overlap. Specifically, for a given source point cloud $\mathcal{P}$ and a target point cloud $\mathcal{Q}$, the ground-truth overlapping mask at the superpoint level, $\mathbf{M}_g^i$, is defined as:

$$\mathbf{M}_g^i = \begin{cases} 1 & \text{if } \hat{\mathcal{P}}_i \text{ correspondent to } \hat{\mathcal{P}}_j \\ 0 & \text{otherwise} \end{cases}, \tag{16}$$

$$\mathbf{p}_t^i = \begin{cases} p_i & \text{if } \mathbf{M}_g^i = 1 \\ 1 - p_i & \text{otherwise} \end{cases}, \tag{17}$$

$$\mathcal{L}_{mask} = \frac{1}{|\mathbf{M}_g|} \sum_i^{|\mathbf{M}_g|} -\alpha(1 - \mathbf{p}_t^i)^\gamma \log(\mathbf{p}_t^i), \tag{18}$$

where $\mathbf{p}_i$ denotes the mask probability of model output. $\mathbf{p}_t^i$ denotes the probability that the mask value is true. $\gamma = 2.0$ and $\alpha = 0.25$ are the focusing parameter and balancing parameter, respectively.

### A.3 PSEUDOCODE OF CROSS3DREG METHOD

Algorithm 1 presents the pseudocode of the proposed Cross3DReg method, detailing its overall process.

### A.4 MORE QUALITATIVE RESULTS

Figure 9 shows additional registration results. Given the unaligned image $\mathbf{I}$, source and target point clouds $\mathcal{P}$ and $\mathcal{Q}$, we visualize the predicted overlapping area, which is the highlighted parts. We also show the visual comparison of the registration results between the proposed Cross3DReg method and ground truth.

### A.5 THE ROBUSTNESS AND GENERALIZATION OF THE CROSS3DREG METHOD

Our method utilises unaligned images to assist predicting the candidate overlap region without calibration information. To evaluate the robustness and generalization of the proposed method, we first

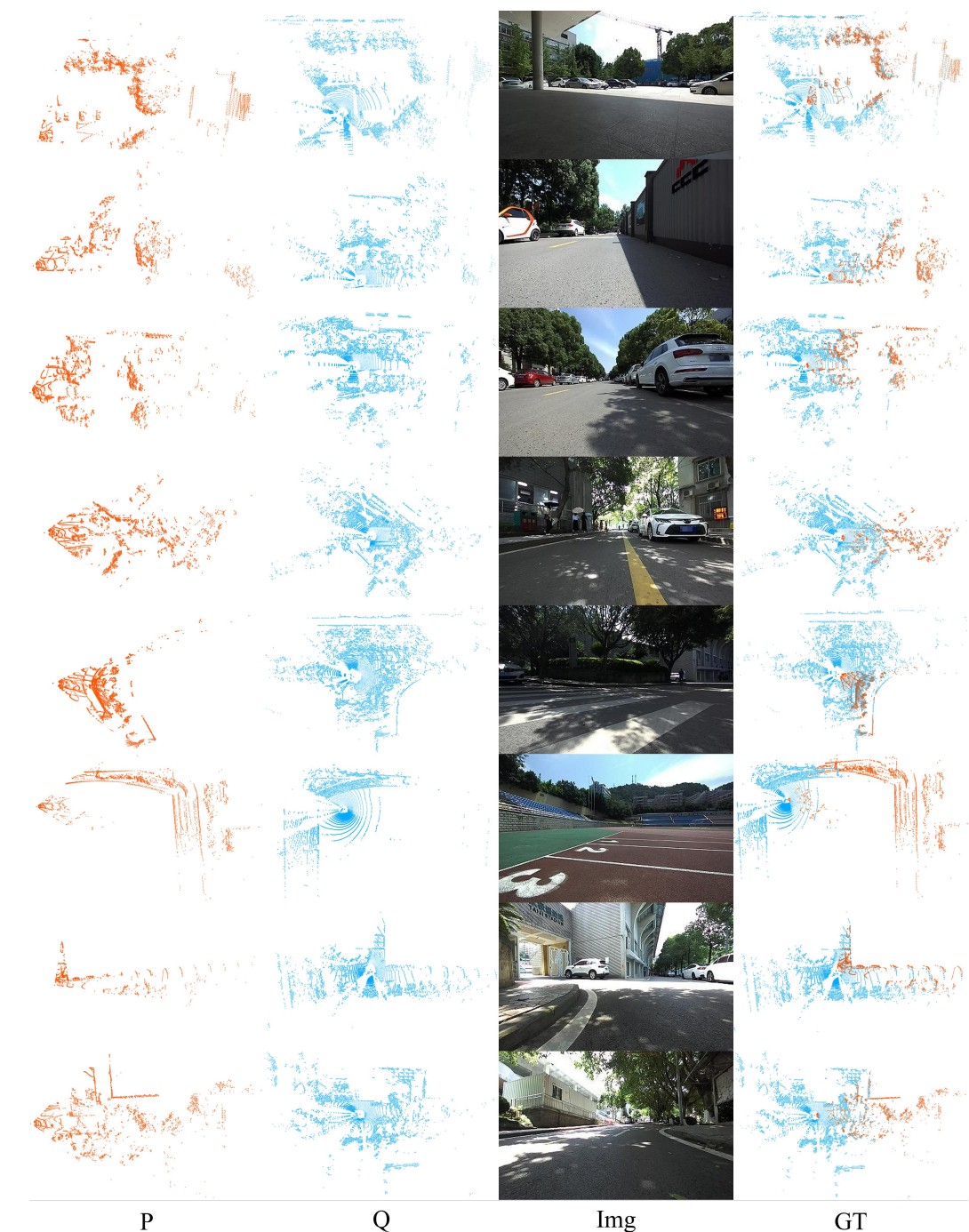

| P | Q | Img | GT |

Figure 8: The visualization of Cross3DReg dataset.

evaluate the registration accuracy when different image sequences are adopted as visual information. As illustrated in Figure 10, we select the original input image alongside frames positioned 20 frames before and after the current frame for comparison. The results demonstrate that our approach maintains excellent stability even when confronted with input images captured from other viewpoints.

We also evaluate the scenarios where images are captured with different installation positions of RGB cameras. Here, we test the registration results when the visual information are obtained with images of the top RGB camera on our autonomous vehicle platform. As we can see in Figure 11, $I_1$ is the original image from ZED camera and $I_2$ is the image from the top RGB camera. Images from the top RGB camera present obvious distortion and some examples are overexposed where

---

**Algorithm 1** Cross3DReg Approach

---

**Input**: source points $\mathcal{P}$, target points $\mathcal{Q}$, unalliged image $\mathcal{I}$.
**Output**: Transfmation matrix $\mathcal{T}$

1: $(\hat{\mathcal{P}}, \mathcal{P}'), (\mathcal{F}_{\hat{\mathcal{P}}}, \mathcal{F}_{\mathcal{P}'}) \leftarrow$ PointFeatureExtractor($\mathcal{P}$)
2: $(\hat{\mathcal{Q}}, \mathcal{Q}'), (\mathcal{F}_{\hat{\mathcal{Q}}}, \mathcal{F}_{\mathcal{Q}'}) \leftarrow$ PointFeatureExtractor($\mathcal{Q}$)
3: $\mathcal{F}^{\mathcal{I}} \leftarrow$ ImgFeatureExtractor($\mathcal{I}$)
4: $\mathbf{P}^{\hat{Q}}_{overlap}, \mathbf{P}^{\hat{P}}_{overlap} \leftarrow$ OMP($\hat{\mathcal{P}}, \mathcal{F}_{\hat{\mathcal{P}}}, \hat{\mathcal{Q}}, \mathcal{F}_{\hat{\mathcal{Q}}}, \mathcal{F}^{\mathcal{I}}$)
4: $\mathbf{M}_{\hat{\mathcal{P}}} \leftarrow (\mathbf{P}^{\hat{P}}_{overlap} > \lambda), \mathbf{M}_{\hat{\mathcal{Q}}} \leftarrow (\mathbf{P}^{\hat{Q}}_{overlap} > \lambda)$
5: $\bar{F}_{\tilde{P}'_i}, \bar{F}_{\tilde{Q}'_j} \leftarrow$ VGAM($\hat{\mathcal{P}}, \mathcal{F}_{\hat{\mathcal{P}}}, \mathbf{M}_{\hat{\mathcal{P}}}, \hat{\mathcal{Q}}, \mathcal{F}_{\hat{\mathcal{Q}}}, \mathbf{M}_{\hat{\mathcal{Q}}}$)
5: **for** $i \leftarrow 1$ to $N$ **do**
5:   **for** $j \leftarrow 1$ to $M$ **do**
5:     $Z'_{ij} \leftarrow \exp\left(-\|\bar{F}_{\tilde{P}'_i} - \bar{F}_{\tilde{Q}'_j}\|^2_2\right)$
5:   **end for**
5: **end for**
6: $\mathcal{C}' \in \{(\tilde{\mathbf{P}}'_i, \tilde{\mathbf{Q}}'_j) | \tilde{\mathbf{P}}'_i \in \hat{\mathcal{P}}, \tilde{\mathbf{Q}}'_j \in \hat{\mathcal{Q}}\} \leftarrow$ select top $K$ $Z'_{ij}$
7: $\mathbf{G}_{\tilde{\mathbf{P}}'}, \mathbf{G}_{\tilde{\mathbf{Q}}'} \leftarrow GroupPints(\mathcal{C}', \mathcal{Q}', \mathcal{P}')$
8: $\mathbf{G}^{\mathbf{F}^{\mathcal{P}'}}, \mathbf{G}^{\mathbf{F}^{\mathcal{Q}'}} \leftarrow GroupFeats(\mathbf{G}_{\tilde{\mathbf{Q}}'}, \mathbf{G}_{\tilde{\mathbf{P}}'}, \mathcal{F}_{\mathcal{Q}'}, \mathcal{F}_{\mathcal{P}'})$
8: **for** $i \leftarrow 1$ to $N_{c'}$ **do**
8:   $\mathbf{S}_i = \frac{\mathbf{G}^{\mathbf{F}^{\mathcal{P}'}}_i (\mathbf{G}^{\mathbf{F}^{\mathcal{Q}'}}_i)^T}{\tilde{d}}$
8: **end for**
9: $C_i \leftarrow$ select top $K'$ in $\mathbf{S}_i$
10: $\mathcal{C}_{fine} \leftarrow C_1 \cup ... \cup C_{N_c}$
11: $\mathcal{T} \leftarrow$ Estimator($\mathcal{C}_{fine}$)
12: **return** $\mathcal{T} = 0$

---

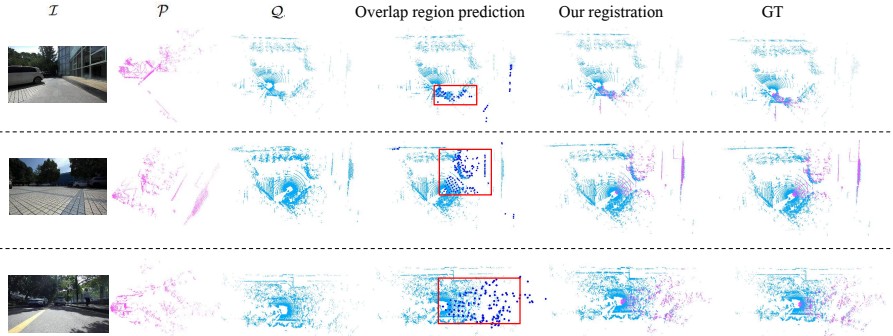

Figure 9: Additional visualization of Cross3dReg registration results.

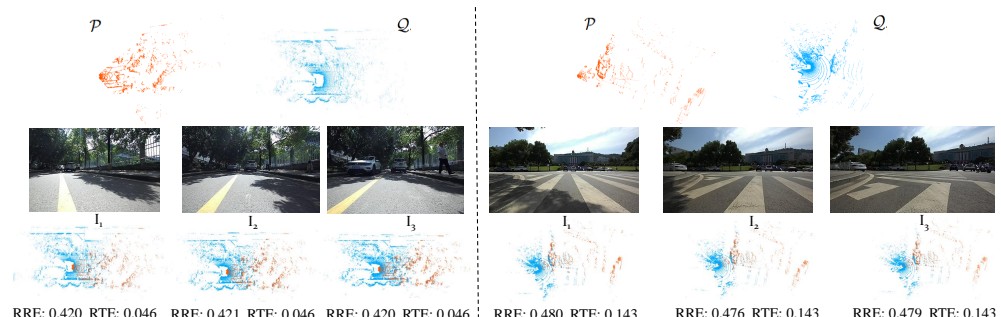

RRE: 0.420 RTE: 0.046    RRE: 0.421 RTE: 0.046    RRE: 0.420 RTE: 0.046    RRE: 0.480 RTE: 0.143    RRE: 0.476 RTE: 0.143    RRE: 0.479 RTE: 0.143

Figure 10: The registration when input images are captured from other views. $\mathbf{I}_1$ represents the image 20 frames prior to $\mathbf{I}_2$, while $\mathbf{I}_3$ denotes the image 20 frames subsequent to $\mathbf{I}_2$.

details cannot be seen. With these challenging and imperfect images, our approach still maintain superior performance in such cases. These experiments verify that our method is robust to different hardware installation settings, low-quality images and easy to generalized to arbitrary camera and LiDAR relative positions.

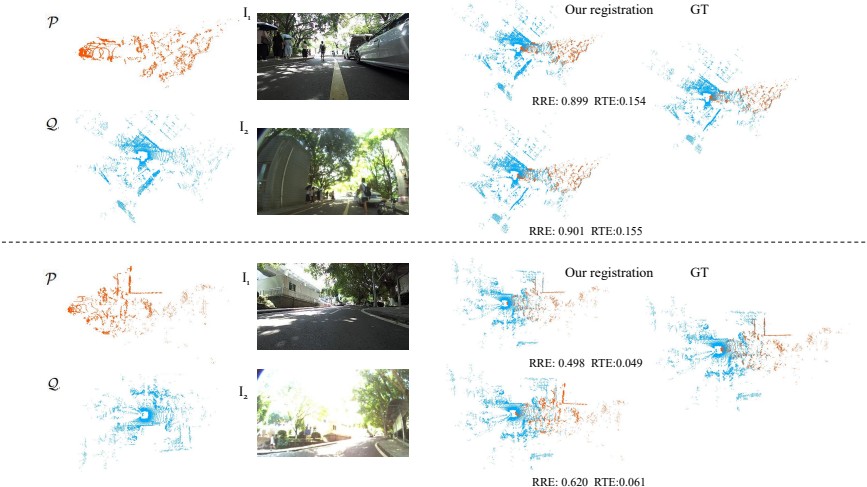

Figure 11: The visualization registration result by using a distorted and overexposed image captured from the RGB camera with different installation position.

Furthermore, Figure 12 displays the registration recall of each method under different relative rotation and translation error thresholds, utilizing different pose estimators. The results demonstrate that when robust pose estimators such as LGR and RANSAC are employed, the poses computed from the correspondences generated by our method exhibit the highest registration recall across all error thresholds.

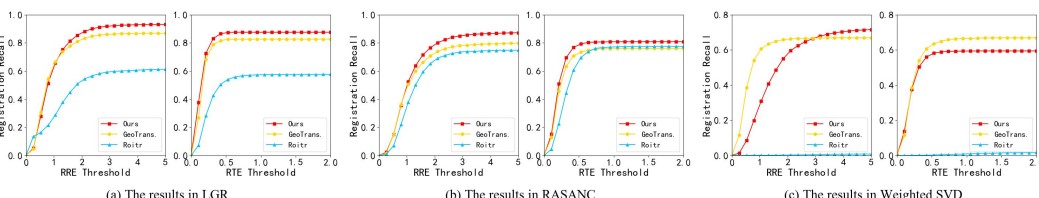

(a) The results in LGR                (b) The results in RASANC                (c) The results in Weighted SVD

Figure 12: Registration recalls with different RRE and RTE thresholds in different estimators on Cross3DReg.