# OpenReview forum: "Cross3DReg: Towards a Large-scale Real-world Cross-source Point Cloud Registration Benchmark"
_ICLR.cc/2026/Conference — ICLR 2026 Conference Withdrawn Submission_

### Official Review · Reviewer_dtjm · 2025-10-22

**Soundness:** 2
**Presentation:** 2
**Contribution:** 3
**Rating:** 4
**Confidence:** 5

**Summary:**

This paper addresses the more specific and challenging problem of cross-source point cloud registration, and proposes a real-world dataset named Cross 3DReg for outdoor scenes, which can be used for training and to some extent fills the gap in the field. At the same time, a cross-source registration framework based on cross-registration is also proposed. It uses unaligned images to predict the overlapping areas of the source point cloud and the target point cloud, effectively filtering out redundant points in the irrelevant areas and significantly reducing the interference of noise in the non-overlapping areas. Finally, cross-source point cloud registration is achieved.

**Strengths:**

1. In the previous research field, either the cross-source datasets were too small to support training, or they were synthetic datasets. The real-world cross-source datasets of this paper have to some extent filled the gap in this field.
2. The article is well-written and easy to understand.
3. This paper uses image features to enhance point cloud features to solve the data problems between cross-source point clouds, which is an effective idea.

**Weaknesses:**

1. Although cross-source point cloud registration is novel, the dataset you provided, Cross 3DReg, consists of rotating mechanical lidar and mixed semi-solid lidar from different sources. I believe this is essentially radar scanning, and the distribution differences between the data of the two are not significant. The research significance and value are not great, and it is questionable.
2. The capitalization of the names of the baselines needs to be standardized, such as RoITr.
3. The superscripts in formulas and symbols (point clouds and point cloud features) are somewhat chaotic and not conducive to reading. It is recommended to unify and standardize them.
4. The method in this paper is an improvement based on the Geotrans framework, adding two modules, OMP and VGAM. The innovation at the method level is not significant.
5. The ablation experiment setting is unreasonable. A set of VGAM w/o OMP experiments should be added in Table 3. Only from (a) and (d), it is not clear whether it is the influence of OMP or VGAM. The effect of OMP is not clear.

**Questions:**

1. Based on experience, it is necessary to re-examine whether the OMP module is necessary. Firstly, in some cases, the overlapping estimation of images and point clouds may introduce incorrect masks. Secondly, the overlapping area between the point clouds selected by the two image overlapping masks is smaller than that between the two original point clouds. This reduction in overlap will definitely affect the registration effect. Thirdly, if the data volume is sufficient, I think directly using visual features to enhance the complete point cloud features instead of applying masks to the point clouds may yield better results.
2. You mentioned in 139-141 that the recent methods Zhang et al. (2022a); Xu et al. (2024; 2025) have introduced other modal information, which may fail in cross-source point cloud registration tasks. However, in your method, you also introduced images (containing color, semantic, and texture information), but you only compared it with the point cloud input methods (such as GeoTrans) in the experiments, which is somewhat unfair. Please add comparative experiments with methods that incorporate multimodal input.

---

> ### Author Response · Authors · 2025-11-23
> **Response to Reviewer dtjm**
>
> We thank the reviewer for the time and effort spent on our research. We also thank the reviewer for recognizing our effective idea, dataset and writing of paper.
>
> For the weaknesses and questions, we respond one by one as follows.
>
> (1) The research significance and value are not great, and it is questionable.
>
> We have compared the point clouds captured with two different lidars in Figure 1 in submission. As we can see, there exist significant distribution difference, pattern difference, resolution difference, and noise and outliers. These differences are caused by fundamental differences in scanning mechanisms and data representation across heterogeneous LiDAR scanners.
>
> Specifically, rotating mechanical LiDAR achieves 360° uniform scanning, producing point clouds with regular distribution, high density, and stable accuracy, which are commonly used in high-precision map construction. In contrast, hybrid semi-solid-state LiDAR results in point clouds that are denser in the center and sparser at the edges due to its scanning way. Semi-solid-state lidar suffers from weaker recognition capabilities when dealing with objects with low reflectivity.
> The point patterns, distributions and levels of noise are sometimes different even when rotating mechanical LiDAR and semi-solid-state LiDAR scan the same objects.
>
> These underlying challenges are the fundamental reason why a large number of recent research focusing on the cross-source point cloud matching, registration, heterogeneous LiDAR place recognition and cross-source localization et al. It is a commonly recognized task.
>
> Also, we evaluate a large amount of current point cloud registration methods, including same-source and cross-source and find that they almost totally fail in most cases. This is because our dataset present challenging scenarios existing in real world. Current publicly available datasets do not show these challenges. Cross3DReg fills the current gap in cross-source point cloud registration community.
>
> (2) The capitalization of the names of the baselines needs to be standardized, such as RoITr.
> (3) The superscripts in formulas and symbols (point clouds and point cloud features) are somewhat chaotic and not conducive to reading. It is recommended to unify and standardize them.
>
> We are sorry for this and revised it in the new version.
>
> (4) The novelty of the method.
>
> Traditional cross-source point cloud registration methods  are only designed for synthetic cross-source settings like Kinect-SFM, not real-world cross-source scenarios. They mainly rely on geometric point cloud features. Facing more challenging scenes presented by Cross3DReg dataset, geometric features based methods cannot address this problem. Image is an important clue to find consistent correspondences between source and target.  However, current multimodal registration methods usually rely on explicit calibration between RGB camera and Lidar. This hinders the practical use of registration methods. The direct fusion of image information with geometry has been proven to introduce much noise and degrade the registration accuracy.
>
>  In contrast, our proposed method is the first method that can effectively utilize unaligned image to achieve accurate cross-source point cloud registration.  It addresses the previous problem of avoiding introduce noise under the condition of unavailable calibration information. It provides a new perspective to use unaligned images for point cloud registration. These improvements make our method of great practical value.
>
> (5) A set of VGAM w/o OMP experiments should be added in Table 3.
>
> We have added the ablation study of VGAM w/o OMP experiment in Table 3. The registration recall is 82.1%. Compared with (b) and (e), it can be seen that the OMP module can effectively mitigate the interference of redundant and noisy points, significantly improving the accuracy of cross-source point cloud alignment.

---

> > ### Author Response · Authors · 2025-11-23
> > **Response to Reviewer dtjm (Continued)**
> >
> > (6) Based on experience, it is necessary to re-examine whether the OMP module is necessary. Firstly, in some cases, the overlapping estimation of images and point clouds may introduce incorrect masks. Secondly, the overlapping area between the point clouds selected by the two image overlapping masks is smaller than that between the two original point clouds. This reduction in overlap will definitely affect the registration effect. Thirdly, if the data volume is sufficient, I think directly using visual features to enhance the complete point cloud features instead of applying masks to the point clouds may yield better results.
> >
> > Based on the added ablation study, OMP module is necessary for accurate point cloud registration. Also, we have to restate that we do not estimate the overlap region of images and point clouds. Image is used to predict the probability of a point belonging to the overlap region between source and target point clouds. Correspondences are matched within these points.  This will not result in the reduction of overlap.
> >
> > As for "the direct use of visual features to enhance the complete point cloud features may yield better results", we also evaluate IMFNET method which directly fuse visual features with point cloud features by an attention mechanism. The experimental results are as follows: RRE=94.8, RTE=9.07, RR=0%. This is because that the direct fusion of visual feature to point clouds has been proven to introduce extra noise to features by previous works, like IGReg, S^2Reg and usually degrades the registration performance. IMFNET have been added to Table 1 in the revised version.
> >
> > (7) Please add comparative experiments with methods that incorporate multimodal input.
> >
> > Previous multimodal methods like Zhang et al. (2022a); Xu et al. (2024; 2025) , rely on explicit calibration information to avoid introduce noise when fusing image. However, in Cross3Dreg dataset, for the sake of practical use, the calibration of camera and LiDAR are unavailable. Thus, these multimodal methods are infeasible for the proposed cross-source point cloud registration scenario. For a fair comparison, we add additional multimodal method, i.e. IMFNET, which incorporates image and point cloud input. We have included this in the revised manuscript.

---

> ### Comment · Reviewer_dtjm · 2025-11-28
> **Response to the rebuttal**
>
> Thank you for your reply and rebuttal.
>
> Some of my concerns have been addressed, but there are still a few issues remaining.
>
> For response 1:
>
> (a) You evaluated other mainstream point cloud registration methods on your dataset and benchmark. I wonder if you retrained these methods? As shown in Table 1, it's hard for me to believe that Predator and CoFiNet have an RR of 0, but GeoTrans has 81%. Such results are hard to convince. Although GeoTrans has made very effective improvements, these are all algorithms within the same framework, and it's hard to imagine such a large gap.
>
> (b) Also, I would like to know why the RRE is so large, over 100. Shouldn't the RRE calculation metric only consider the successfully registered samples?
>
>
> For response 6:
>
> (a) This probability will definitely affect the possibility of matching at this point. I still think it will affect the overlapping area. Could you further explain the reason why you said "This will not result in the reduction of overlap"?
>
> (b) I'm not quite clear. "This is because that the direct fusion of visual features to point clouds has been proven to introduce extra noise." Why would this kind of feature fusion introduce noise?

---

> ### Author Response · Authors · 2025-12-01
> **Response to dtjm**
>
> Thanks for your response.
>
> (a) You evaluated other mainstream point cloud registration methods on your dataset and benchmark. I wonder if you retrained these methods? As shown in Table 1, it's hard for me to believe that Predator and CoFiNet have an RR of 0, but GeoTrans has 81%. Such results are hard to convince. Although GeoTrans has made very effective improvements, these are all algorithms within the same framework, and it's hard to imagine such a large gap.
>
> Response:
> - All mainstream point cloud registration baselines, except ICP, were retrained on the Cross3D training set and then evaluated on the test set.
>
> - The analysis of the performance:
>
> We attribute these results to three primary factors:
>
> First, the inherent challenge of the dataset. Our dataset presents challenges like differences in structural pattern and density, and realistic noise and outliers presented in real-world cross-source point clouds. These challenges make the geometry information broken and the geometric features are unreliable.
>
> Second, the lack of consistent information for enhancement in existing methods: Keypoint-based methods (e.g., Predator, FCGF): Although relying on correspondences, the combined impact of low overlap and cross-source interference (modal discrepancies, density variations, and noise) significantly degrades feature consistency. Specifically, Predator fails to accurately predict pointwise overlap scores, leading to the erroneous extraction of correspondences from incorrect (non-overlapping) regions. Keypoint-free methods (e.g., CoFiNet, RoITr, GeoTransformer): These methods employ a "coarse-to-fine" strategy, where the accuracy of the initial super-point matching is pivotal. However, under the influence of modal discrepancies and noise, super-point feature extraction becomes unreliable without the guidance of consistent features. Consequently, Geotrans and RoITr leverage geometric structural features (angles and distances between super-points) and rotation-invariant features (PPF), respectively. These mechanisms significantly enhance super-point feature consistency and matching accuracy, yielding reliable fine-grained correspondences. In contrast, our experiments reveal that CoFiNet exhibits weak super-point feature consistency and lacks the guidance of such consistent features, leading to numerous mismatches in the coarse stage that propagate to the fine-grained stage. As illustrated in Figure 4, Predator and CoFiNet generate a substantial number of outliers. These erroneous correspondences directly mislead the pose estimator into computing inaccurate transformation matrices, resulting in significant registration errors.
>
> Finally, the impact of the pose estimator. The choice of back-end optimization also plays a critical role. CoFiNet employs the traditional RANSAC pose estimator, whereas Geotrans utilises the Local Global Registration (LGR) estimator. The key difference lies in their approach: LGR computes local transformation matrices using correspondences within local patches via Weighted SVD, followed by a global voting scheme to select the optimal transformation for iterative refinement. This process circumvents the tens of thousands of random sampling iterations required by RANSAC. In our tests, LGR achieved a nearly 100x speedup compared to RANSAC while maintaining, or even exceeding, registration accuracy. Please refer to Table 3 for detailed comparative experimental data with different pose estimators.
>
> (b) Also, I would like to know why the RRE is so large, over 100. Shouldn't the RRE calculation metric only consider the successfully registered samples?
>
> Response: In all experiments, we follow the evaluation established by Geotrans and CoFiNet, reporting the average RRE and RTE calculated across all point cloud pairs in the test set. We argue that calculating error metrics based solely on successfully registered samples introduces a selection bias; such results are inherently bounded by the thresholds used to define Registration Recall (RR). Consequently, we believe that this artificially lowers the reported error and fails to provide a fair or holistic assessment of a method's actual registration precision in practical scenarios. Therefore, we report the average over the entire test set to ensure a rigorous comparison.

---

> > ### Author Response · Authors · 2025-12-01
> > **Response to dtjm (Continued)**
> >
> > (c) Could you further explain the reason why you said "This will not result in the reduction of overlap"?
> >
> > Response: Our method employs the OMP module to predict the mask for points to see if the point is likely to fall within the image's field of view. Points within the image views are used to locate correspondences between source and target. These correspondences form the overlap region.
> >
> > (d) I'm not quite clear. "This is because that the direct fusion of visual features to point clouds has been proven to introduce extra noise." Why would this kind of feature fusion introduce noise?
> >
> > Response: In the absence of camera calibration, IMFNET adopts a strategy of fusing image features directly into the global point cloud. This often leads to incorrect correspondences during keypoint matching, which in turn degrades the accuracy of pose estimation. Direct fusion on the global point cloud erroneously assigns image features to a large number of irrelevant points. In contrast, calibration-based methods, such as IGREG and PCR-REG, leverage calibration parameters to establish precise point-to-pixel correspondences. This facilitates the accurate mapping of image features onto the point cloud, thereby enhancing the discriminability between valid keypoints and irrelevant points and preventing erroneous correspondences.
> >
> > We thank the reviewer again and revised the corresponding parts in the revised manuscript.

---

### Official Review · Reviewer_5DW7 · 2025-10-26

**Soundness:** 2
**Presentation:** 2
**Contribution:** 2
**Rating:** 2
**Confidence:** 4

**Summary:**

This paper focuses on the task of cross-source point cloud registration, which aims to align 3D data collected from different types of sensors. Compared with same-source registration, this task is more challenging due to the lack of large-scale real-world datasets and the inherent differences in sensor characteristics. To address these issues, the authors construct Cross3DReg, a large-scale real-world multi-modal dataset collected using a rotating mechanical LiDAR and a hybrid semi-solid-state LiDAR. They further propose an overlap-based registration framework that predicts overlapping regions between unaligned images and point clouds to filter out irrelevant points, and introduce a visual-geometric attention module to enhance feature consistency across modalities.

**Strengths:**

This paper addresses the problem of cross-source point cloud registration, which is technically challenging.

The construction of a cross-source registration dataset represents a contribution to the community.

The manuscript is overall well written and easy to follow.

**Weaknesses:**

1. The core concern lies in the introduction of additional image information during the point cloud registration process. This raises questions about the fairness of the comparison for the baselines do not use the images, and Table 3 shows that the performance improvement primarily stems from the inclusion of image data.

3. Although the authors claim that the images and point clouds are not geometrically aligned, they still assume a significant spatial overlap between the image and the two point clouds. Moreover, the registration process is conducted only within these “image-pc overlapping regions”, which seems difficult to achieve in practice.

3. Furthermore, first determining the overlapping regions between the image and point clouds appears somewhat counter-intuitive, since the modality gap between images and point clouds is typically larger than that between cross-source point clouds.

**Questions:**

Please refer to the weaknesses listed above.

---

> ### Author Response · Authors · 2025-11-23
> **response to Reviewer 5DW7**
>
> We thank the reviewer for the time and effort spent on our research.
>
> For the weaknesses and questions, we respond one by one as follows.
>
> (1) The fairness of the comparison for the baselines without using images.
>
> Please refer to the above response to Reviewer 3obY for the first question.
>
> (2) Table 3 shows that the performance improvement primarily stems from the inclusion of image data.
>
> Table 3 shows the ablation study of our method.  Experiments are set up with five scenarios in comparison: (a) only the geometric self-attention module is used; (b) only the VGAM is used; (c) the OMP module is used with the vanilla attention module; (d) the OMP module is used with the geometric self-attention module; and (e) the complete Cross3DReg method. The comparison among (a), (b), (e) shows that the OMP module can effectively mitigate the interference of redundant and noisy points, significantly improving the accuracy of cross-source point cloud alignment. The comparison of schemes (b), (c), (d), and (e) further shows that our visual-geometric feature attention guidance module improves the feature consistency among cross-source point clouds, enhancing the registration accuracy. In general, OMP uses images to help locate possible points belonging to the overlap region between source and target point clouds without calibration. VGAM module utilises image information to enhance the feature consistency, allowing for more accurate correspondences. These two modules work together. To be clearer, we have refined the ablation study part. Please refer to Page 10.
>
> (3) Although the authors claim that the images and point clouds are not geometrically aligned, they still assume a significant spatial overlap between the image and the two point clouds. Moreover, the registration process is conducted only within these “image-pc overlapping regions”, which seems difficult to achieve in practice.
>
> We do not locate the“image-pc overlapping regions”. In our research, we use one image to guide the overlap region estimation between source and target point clouds.  OMP module predicts the masks of points to see if the point is within the image view or not. It is reasonable that only points within the image view are possible in the overlap region between the source and target point clouds. Both the source and target point clouds are processed by the OMP module to acquire points that are possible in the overlapping area. Correspondences are matched within these points.  The VGAM module is designed with the consideration that the overlap region between point clouds should be feature-consistent. However, due to the different point distribution, noise and outliers in cross-source point clouds, the geometric feature is unreliable. Thus, an image is regarded as an effective bridge to enhance feature consistency. The images are not required to be aligned with the point clouds. Only one image captured in the same scenario is needed. Extensive experiments in the submission have shown that our method is accurate under conditions of images prior to/after source and target point clouds or various low-quality images in Figures 10 and 11. Thus, compared with current multimodal point cloud registration methods, our approach operates under very relaxed constraints regarding image input. These advantages show that our method is of strong practical value.
>
> (3) First determining the overlapping regions between the image and point clouds appears somewhat counter-intuitive, since the modality gap between images and point clouds is typically larger than that between cross-source point clouds.
>
> Please refer to the above question.

---

> > ### Comment · Reviewer_5DW7 · 2025-11-26
> > **Response to the rebuttal**
> >
> > Thank you for the rebuttal. However, I still have reservations regarding the core assumption that the image view covers the overlap region.
> >
> > In rebuttal, the authors state that "It is reasonable that only points within the image view are possible in the overlap region between the source and target point clouds. " and this is the key assumption of this paper. But this does not make that sense for me.
> >
> > In point cloud registration task, the overlap region is unknown a prior. If we rely on an image to bridge the source and target, how can we guarantee that this image captures their exact overlap area without knowing the relative pose beforehand? If the image is captured randomly and misses the overlap region, does the method fail (I think it fails because the following registration is conducted in the ``image-to-pc overlapping regions'')? This assumption seems to limit the method's applicability in unconstrained scenarios.
> >
> > For the reviewer, prior works that rely on aligned image–point-cloud pairs operate under the realistic condition of having calibrated multimodal data. In contrast, this method assumes that the image does not need to be aligned with the point clouds, yet it must still capture the overlapping region between two unaligned point clouds. This combination of assumptions makes me uncertain about the applicability of the method in real-world scenarios.

---

> ### Author Response · Authors · 2025-11-27
> **response to Reviewer 5DW7**
>
> We thank the reviewer for the time and effort.
>
> -  In point cloud registration task, the overlap region is unknown a prior. If we rely on an image to bridge the source and target, how can we guarantee that this image captures their exact overlap area without knowing the relative pose beforehand?
>
> Response: We think it is better to clarify the definition of the overlap region here. The overlap region is composed of a set of correspondences. In our work, images are assumed to be captured from a rough co-located viewpoint. It provides a clue to find the overlap between the source and the target as images present consistent visual information. In the OMP module, it first fuses visual and geometric features and then is trained to predict the mask of a source/target point. Those points whose mask=1 in the source and target will be used to locate correspondences. These correspondences form the overlap region between source and target.  Please note that in the training stage, the ground-truth poses between source and target are known and used to train our registration model.
>
> - The applicability of the method in real-world scenarios.
>
> Response: Actually, in most of the current multimodal point cloud registration datasets, like 3DMatch and 3DLoMatch, the views of images cover most of the point clouds, including the overlap regions. All the multimodal point cloud registration methods are based on the assumption that images can provide effective information to help registration.
>
>  Previously, the calibration information must be known to fuse the image and geometric features to avoid introducing noise. This means the image and point clouds must be captured with a calibrated setup. However, in some actual applications, it is often the case that the calibration is inaccurate because of the inevitable vibration of the Lidar during usage and noise is caused. The distortion of images and the low resolution of Lidar will also degrade the calibration. Also, there is a crucial yet often overlooked point is the extrinsic parameter drift. Even if precisely calibrated at the time of manufacture, the extrinsic parameters can change over time.
>
> In practical applications, based on our method, the camera can capture an image from a roughly co-located viewpoint, even if the image quality is suboptimal (Our experiments have demonstrated that even distorted and overexposed images can help registration in our method). It is also not required that the images be captured at the same time with point clouds. It can be captured offline. In contrast, the calibration-based methods require the camera and LiDAR to be rigidly fixed, necessitating frequent and precise recalibration to maintain accurate pixel-to-point correspondences.  It is obvious that our method is more practical than calibration-based methods.

---

> ### Comment · Reviewer_5DW7 · 2025-11-28
> **Response to the rebuttal**
>
> Thank you for the rebuttal.
>
> I think the explanation that "images are acquired approximately synchronously with the hybrid solid-state LiDAR, but inevitable vibration and odometry drift render calibration unavailable" is more direct and convincing than the original text. As a result, my core issue regarding the feasibility of the setup is resolved.
>
> However, I can only raise my score to 4 at this stage (It seems I cannot edit right now; I need to wait for the edit button to be available.), as I think the paper still needs some revisions:
>
> 1. The main text needs to explicitly state that "the images are nearly collected at co-located view (synchronously or else) with the hybrid solid-state LiDAR but contain unknown offsets.", sounds like an uncalibrated RGB-D setup.  Although Line 245 mentions a "common view," the phrasing in Lines 90 and 251 implies a level of randomness that still raises doubts.
>
> - I think roughly aligning the image and semi-state LiDAR will not negatively impact your claim of "very relaxed constraints regarding image input" (as "very relaxed" does not imply "no constraints"). Nor does it conflict with the usage setting that data 'can be captured offline, but requires co-visibility'. On the contrary, it makes the methodology more convincing.
> - Without this constraint, readers might imagine a failure case: Suppose one attempts to register the hybrid solid-state LiDAR (HL) at time $ T $ with the 64-beam LiDAR (64L) at time $ T+t $, where the HL might only overlap with the back part of the 64L, but if I randomly use the image at time $ T+t $ only covers the front view of the 64L, the image would fail to provide any positive guidance. The text must clarify that the setup avoids such disjointed overlap.
>
> 2. As a Datasets and Benchmarks area submission, the dataset description should not mainly appear in the Appendix.
> I suggest moving key details (e.g., Table 4) to the main text.
> This section should also explicitly describe the data acquisition setting mentioned in Point 1.

---

> > ### Author Response · Authors · 2025-11-30
> > **Response to Reviewer 5DW7**
> >
> > Thanks for your useful advice. We addressed all the comments and revised the corresponding parts in the updated manuscript.
> > Thanks again for your time and effort.

---

### Official Review · Reviewer_XRz2 · 2025-10-31

**Soundness:** 2
**Presentation:** 2
**Contribution:** 2
**Rating:** 4
**Confidence:** 3

**Summary:**

This paper proposes a cross-source point cloud registration method that aligns point cloud data from different sensors. The authors construct a large-scale multi-modal dataset for cross-source point cloud registration. They introduce an overlap-based registration framework that uses unaligned images to predict overlapping regions between source and target point clouds. Additionally, they propose a visual-geometric attention-guided matching module that enhances cross-source point cloud feature consistency by fusing image and geometric information.

**Strengths:**

- A large scale cross source point cloud registration dataset is constructed.
- An overlap-based cross-source point cloud registration method is proposed.

**Weaknesses:**

- The dataset construction uses a hybrid semi-solid-state LiDAR to capture source point clouds and a 64-line rotating mechanical LiDAR for target point clouds. However, the rationale for this choice is unclear. Is this a common practice in real-world applications?
- One key to the proposed method's success is predicting the overlap region between source and target point clouds, as well as between point clouds and images. More experimental results demonstrating this prediction should be added.
- The experiments are insufficient. It would better to also evaluate the proposed method on other cross-source point cloud registration datasets. In addition, more state-of-the-art cross-source point cloud registration methods discussed in related work should be compared.

**Questions:**

- What is the rationale for constructing the cross-source point cloud registration dataset in such a way?
- How does the proposed method perform on other existing cross-source point cloud registration datasets?

---

> ### Author Response · Authors · 2025-11-22
> **response to  Reviewer XRz2**
>
> We thank the reviewer for the time spent on our research.
>
> For the weaknesses and questions, we respond one by one as follows.
>
> (1)the rationale for constructing the cross-source point cloud registration dataset in such a way.
>
> The field of point cloud acquisition has advanced rapidly in recent years.  The rotating mechanical LiDAR achieves 360° uniform scanning, producing point clouds with regular distribution, high density, and stable accuracy, which are commonly used in high-precision map construction. In contrast, hybrid semi-solid-state LiDAR usually results in point clouds that are denser in the center and sparser at the edges due to its scanning way. Semi-solid-state lidar suffers from weaker recognition capabilities when dealing with objects with low reflectivity. The point patterns, distributions and levels of noise are sometimes different even when rotating mechanical LiDAR and semi-solid-state LiDAR scan the same objects. The rotating mechanical LiDAR is usually more expensive and can be used to construct the high-precision map. Semi-solid-state lidar is usually cheaper and typically equipped on small unmanned mobile devices.
> Due to these characteristics, a hybrid semi-solid-state LiDAR for the source point cloud and a rotating mechanical LiDAR for target point clouds play an important role in heterogeneous LiDAR place recognition[1][2], map localization for an embodied agent in diverse degenerate scenarios[3].
>
> [1] Liang, J., Deng, Z., Zhou, Z., Sun, M., Ghasemalizadeh, O., Kuo, C.H., Sen, A. and Manocha, D., 2025. CSCPR: Cross-source-context indoor RGB-D place recognition. IEEE Robotics and Automation Letters.
>
> [2] Jung, M., Jung, S., Gil, H. and Kim, A., 2025. HeLiOS: Heterogeneous LiDAR place recognition via overlap-based learning and local spherical transformer. arXiv preprint arXiv:2501.18943.
>
> [3] Chen, Z., Qi, Y., Feng, D., Zhuang, X., Chen, H., Hu, X., Wu, J., Peng, K. and Lu, P., 2024. Heterogeneous LiDAR dataset for benchmarking robust localization in diverse degenerate scenarios. The International Journal of Robotics Research, p.02783649251344967.
>
> (2) More experimental results demonstrating this prediction of overlap between point clouds and between point clouds and images should be added.
>
> Thank you for pointing this out.  We do locate the overlap region between source and target point clouds but do not find the point-to-pixel correspondences between the point cloud and the image. The image is only used to help find the mask of a point that indicates if a 3D point possibly belongs to the overlap region between source and target point clouds or not. We have clarified and highlighted the Introduction part in the revised manuscript.
>
> As for the  overlap region between point clouds, more results are shown in Figure 5 and Figure 9 in the revised version. Please refer to Page 10 and 17 of the revised manuscript.
>
> (3) The experiments are insufficient. It would better to also evaluate the proposed method on other cross-source point cloud registration datasets. In addition, more state-of-the-art cross-source point cloud registration methods discussed in related work should be compared.
>
> Currently, the main publicly available cross-source point cloud datasets  are 3DCGS and Kitti-CrossSource.  3DCGS contains 202 pairs of indoor LiDAR and SFM cross-source point clouds, while Kitti-CrossSource includes 2006 pairs of outdoor LiDAR and SFM cross-source point clouds. However, neither of these datasets provides image data, making it difficult to validate our proposed method. To fill this gap, we construct a cross-source dataset Cross3DReg, which fills the current gap in multimodal datasets for cross-source point cloud registration. It also provides support for validating the effectiveness of cross-source methods.
>
> Additionally, we have referenced methods such as Cross-PCR, SPEAL, and VRCHF in the related work section. It should be noted that Cross-PCR and SPEAL are not yet open-sourced (Cross-PCR repository: https://github.com/GuiyuZhao/Cross-PCR, SPEAL repository: https://github.com/kezheng1204/KITTI-CrossSource). For more comprehensive comparison, we add a new comparison with a multimodal point cloud registration method, i.e. IMFNET in Table 1. Please refer to Table 1 on Page 8.

---

### Official Review · Reviewer_3obY · 2025-11-01

**Soundness:** 3
**Presentation:** 3
**Contribution:** 3
**Rating:** 6
**Confidence:** 4

**Summary:**

This paper proposes a novel method for cross-source point cloud registration. The key insight is leveraging images to detect overlap regions between point clouds captured by different sensors, thereby addressing registration failures caused by sensor discrepancies. To facilitate research in this area, the authors introduce Cross3DReg, a new large-scale real-world dataset for cross-source point cloud registration, which was previously lacking in the field. Experimental results demonstrate that the proposed method significantly outperforms existing point cloud registration approaches, achieving substantial improvements across evaluation metrics.

**Strengths:**

- **Clear presentation**: The paper is well-written and easy to follow, providing sufficient context to understand both the field and the proposed method's contributions.
- **Valuable dataset contribution**: The proposed Cross3DReg dataset addresses a critical gap in the community, providing the large-scale real-world benchmark that researchers have been seeking for cross-source point cloud registration.
- **Strong experimental validation**: The proposed method demonstrates substantial performance gains over existing baselines. The ablation study effectively validates the contribution of each component in the proposed approach.

**Weaknesses:**

- **Concerns regarding evaluation fairness**: While leveraging auxiliary image data to identify overlap regions and guide point matching through the visual-geometric attention module is a distinctive feature of the proposed method, this additional modality was not available to baseline methods. This asymmetry in input data raises concerns about fair comparison—the performance gains may partially stem from access to richer input rather than solely from algorithmic improvements.

**Questions:**

- **Performance of VRHCF on Cross3DReg**: VRHCF is the only dedicated cross-source point cloud registration method included in the comparison, yet it appears to fail completely on the Cross3DReg benchmark. Could the authors provide insight into why this occurs? Is this due to specific characteristics of the Cross3DReg dataset (e.g., scale, sensor types, scene complexity), limitations in VRHCF's architecture, or implementation issues?

---

> ### Author Response · Authors · 2025-11-22
> **response to Reviewer 3obY**
>
> Many thanks for highly recognising our research with respect to clear presentation, valuable dataset contribution and strong experimental validation.
>
> For the weakness and questions, we respond as follows.
>
> (1)Concerns regarding evaluation fairness. According to our investigation, representative multimodal point cloud registration methods in recent years for point cloud registration include IGREG[1], PCR-CG[2], IMFNET[3], and HybridFusion[4].  These methods primarily focus on the same-source point cloud registration and can be categorised into two groups based on their reliance on camera calibration:
>
>  Calibration-based fusion methods such as IGREG[1], PCR-CG[2], and HybridFusion[4], which leverage intrinsic and extrinsic camera parameters to achieve point-to-pixel projection and then precise image information fusion, effectively improving registration accuracy.
>
> Fusion methods without calibration, like IMFNET[3], which adopt a global fusion strategy through cross-attention mechanisms to embed image features holistically into point cloud representations.
> The calibration-based methods require the extrinsic and intrinsic metrices between cameras and lidars to perform the image-point cloud alignment. However, the acquisition of calibration is usually not straightforward in most cases, and this hinders them in real applications. Considering more general situations in real applications, the Cross3DReg dataset does not provide the calibration information. Calibration-based methods cannot run on the Cross3DReg dataset.
>
> For a fair comparison of multimodal methods, we further evaluate IMFNET on the Cross3DReg dataset. Under identical evaluation conditions, it achieves an RRE of 94.8, RTE of 9.07, and RR of 0.0. These results indicate that naively globally fusing image information may introduce a great amount of noise due to modal differences, interfering with point cloud feature matching and making it difficult to obtain reliable correspondences, ultimately compromising registration accuracy.
>
> Compared with current multimodal methods, the proposed method can achieve high accuracy and, meanwhile, does not require calibration information, enhancing the applications in real scenarios. The experiments of IMFNET demonstrate that the performance gains stem from the algorithm design rather than richer input. We have included the comparison with IMFNET in the revised manuscript. Please refer to Table 1 on Page 8.
>
> (2) Performance of VRHCF on Cross3DReg.
> We have investigated current cross-source point cloud registration methods, and only VRCHF is open-sourced. That's why VRHCF  is the only cross-source method we can evaluate on our dataset.
>
> VRCHF first adopts a spherical voxelization operation to resample the density-variant cross-source point clouds into evenly distributed point clouds and then uses a deep learning method to extract features. A correspondence filtering is performed to remove wrong matches before registration. VRCHF is naturally a geometric feature based point cloud registration method and faces indoor scenes where point clouds are captured with Kinect and synthesised with images. The difference in point distribution between Kinect captured and image-synthesised point clouds are relatively small. It is possible to get  evenly distributed points by downsampling/resampling operations. Also, the levels of noise and outliers are not complex enough in indoor point clouds. Therefore, purely geometric features possibly work in these scenes. However, faced with our Cross3DReg, which fully presents real-world point uneven distribution, noise, outliers, and pattern difference, VRCHF  cannot get evenly distributed point clouds only with a simple sampling method. The extracted geometric feature cannot keep consistency and totally failed in our dataset.
>
> References:
> [1]Xu, Zongyi, et al. "Igreg: Image-geometry-assisted point cloud registration via selective correlation fusion." IEEE Transactions on Multimedia 26 (2024): 7475-7489.
>
> [2]Zhang, Yu, et al. "Pcr-cg: Point cloud registration via deep explicit color and geometry." European conference on computer vision. Cham: Springer Nature Switzerland, 2022.
>
> [3]Huang, Xiaoshui, et al. "IMFNet: Interpretable multimodal fusion for point cloud registration." IEEE Robotics and Automation Letters 7.4 (2022): 12323-12330.
>
> [4]Wang, Yu, et al. "HybridFusion: LiDAR and vision cross-source point cloud fusion." IEEE Robotics and Automation Letters 9.2 (2023): 1139-1146.

---

### Note · Authors · 2026-01-04

I have read and agree with the venue's withdrawal policy on behalf of myself and my co-authors.